# Seismic Response of Aeolian Sand High Embankment Slopes in Shaking Table Tests

**Zhijun Zhou [1], Jiangtao Lei [1] , Shaobo Shi [1] and Tong Liu [2],***

[1] School of Highway, Chang'an University, Xi'an 710064, China; zhouzhijun@chd.edu.cn (Z.Z.); leijiangtao@chd.edu.cn (J.L.); shishaobo@chd.edu.cn (S.S.)

[2] School of Science, Xi'an University of Architecture and Technology, Xi'an 710055, China

* Correspondence: Liutong@xauat.edu.cn

**Abstract:** Aeolian sand high embankments are always damaged by earthquakes; however, little research has addressed this so far. In this study, shaking table tests were conducted on three aeolian sand high embankment models. Based on the shear failure mechanism of aeolian sand, the seismic responses of model embankments were analyzed. When seismic waves were inputted, the horizontal acceleration magnification (HAM) of three models always exceeded 1.0, and showed an increasing trend with height. Furthermore, according to the HAM change rules of three models under different input peak accelerations, the destruction of model embankments under earthquakes includes three stages: the reflected wave emergence (RWE) stage, the reflected wave strengthening (RWS) stage, and the acceleration magnification attenuation (AMA) stage. According to this definition, models with slopes of 1/1.2 and 1/0.8 experienced all three stages during tests, and the critical horizontal acceleration transform from the RWS stage to the AMA stage appeared. The model with a slope of 1/1.5 only experienced RWE and RWS stages during the test. At the end of the tests, the macroscopic instability mechanisms of all three models were studied, which were found to match the distribution law of HAM during tests and the destruction stage definition.

**Keywords:** seismic response; aeolian sand high embankment; shaking table test; instability mechanism

## 1. Introduction

Deserts and sandy soil cover approximately 13% of China's land mass. To develop the economy and exploit mineral resources, it is imperative to develop the transportation network of this desert area. The desert expressway features a small fill volume, which can ensure the stability of the expressway structure and decrease costs. However, in this area with its particular terrain, a large fill volume is inevitable [1,2].

It has been reported that the number of earthquakes in the desert regions of China have increased in recent years. Although most of these are small earthquakes that do not affect people's daily life, long-term sustained and high frequency small earthquakes can have unpredictable effects such as soil loosening, house collapse, and embankment deformation [3–6]. Several desert highways in this area have been damaged to varying degrees due to frequent earthquakes, among which, the damage of high embankments is the most severe [7]. Therefore, it is critical to study the seismic response of aeolian sand high embankment slopes. The distribution of important deserts in China and the location of the relevant research is shown in Figure 1.

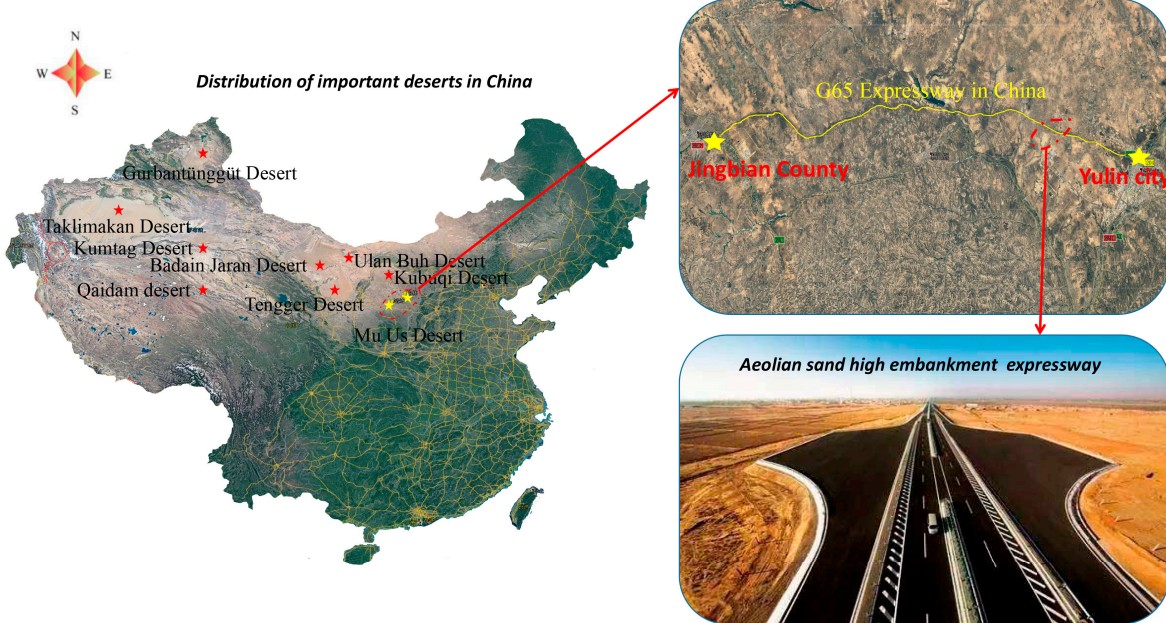

**Figure 1.** Location of this research.

Traditionally, the pseudo static analysis method has been used to analyze the seismic stability of embankments [8–10]. It is widely used for seismic design in current engineering specifications due to its brevity; however, with regard to cases where the stiffness of soil is obviously reduced or when liquefaction occurs during earthquakes, this method is not applicable. Newmark [11] (1965) presented an analysis method by computing permanent displacement, studying the relationship between permanent displacement and critical acceleration. This has been named the Newmark's sliding block method. This method is well known and has been improved by many scholars [12–17]; however, the Newmark's sliding block method still includes rather crude pseudo static analysis assumptions, and is not suitable to study the deformation of soil with the maximum displacement at the centimeter level [18]. Numerical methods—such as the finite element method, Lagrangian element method, and DEM (discrete element method)—were also used by many scholars to study the seismic response of geotechnical structures. Many advanced constitutive models have been adopted and improved by scholars when using numerical methods [19,20]. Examples are, the nonlinear viscoelastic model, th elastic–plastic model, and the boundary surface model [21,22]. However, the finite element method and the Lagrangian element method are well-known methodologies for solving continuum problems exploiting a discrete approach [23]; therefore, they cannot be used to study the seismic response of aeolian sand high embankments with their poor continuity material. The DEM is particularly suitable for stress analysis of joint rock mass. Moreover, it can calculate the deformation and stress distribution of the soil, and it is the most appropriate numerical analysis method for the study of the seismic response of embankments. However, if there are too many particles and actual contacts during simulation, both the calculation speed and accuracy will decrease; therefore, it is not suitable for the simulation of high embankments.

It is possible to study the dynamic response of the slope in laboratories under actual stress conditions due to the development of dynamic centrifuge tests and shaking table tests. These tests have been widely adopted by scholars to investigate the seismic response of embankments [24–26]. The gravitational field can be altered as expected in dynamic centrifuge tests so that the test model can better cater to the requirements of similarity relation [27–30]. However, the carrying capacity of a dynamic centrifuge is extremely limited [31,32], while the shaking table system can be implemented for large-scale model tests with various loading frequencies [33]. The seismic response of the models can be analyzed in depth and the failure mode of models can be observed intuitively [34–36]. Previous studies

on seismic response of embankments mainly focused on soil slope, and these studies were always interested in investigating the seismic response laws of model embankment with different reinforcement measures; however, the failure mechanisms were not described systematically although they seem very promising [37].

In this work, shaking table tests were performed on three aeolian sand high embankment models (i.e., embankment with slopes of 1/1.5, 1/1.2, and 1/0.8, respectively). El-Centro motions, Lan Zhou motions, and sinusoidal waves were used to investigate the horizontal acceleration response of aeolian sand high embankments. By combining the response laws with the shear failure mechanism of Aeolian sand, the damage of aeolian sand high embankments under earthquakes was defined in three stages. Then, at the end of the tests, the macroscopic instability mechanism of these three models was studied.

## 2. Shaking Table Tests

### 2.1. Mechanical Properties and Microcosmic Shear Failure Mechanism of Prototype Sand

The prototype sand was taken from the Mu Us Desert in China. It has been blown by wind for a long time; therefore, sand particles are fine, contain little silt and clay, and most of the particle sizes are in the range of 0.074–0.25 mm. It shows good water permeability, i.e., the sand surface shows almost no physical adsorption of water, and the maximum water absorption is less than 1%. These characteristics result in the non-plastic feature of Aeolian sand, which means that the molding of sand is difficult and the shear strength is poor even after forming. The physical characteristics of prototype sand are as follows according to the laboratory tests: initial density $\rho = 1.59$ g/cm$^3$, uniformity coefficient $C_u = 1.68$, coefficient of curvature $C_c = 0.96$, internal friction angle $\varphi = 37.5^\circ$, maximum dry density $\rho_d = 1.75$ g/cm$^3$, and optimum water content $\omega = 12.36\%$. According to the Test Methods of Soils for Highway Engineering (JTG E40-2007) [38], the prototype sand belongs to fine and poorly graded sand.

The seismic capability of aeolian sand high embankments mainly depends on the occlusion force between sand particles. The micro-failure of the Aeolian sand under horizontal accelerations can be described as a shear-trituration between sand particles, which leads to the generation of cracks. The failure mechanism of prototype sand is shown in Figure 2. Therefore, the shear strength of Aeolian sand plays an important role for the study of seismic performance of Aeolian sand embankment slopes, and shear modulus is a significant parameter for estimating the shear strain resistance of materials. Furthermore, according to the principle of soil dynamics, a functional relationship exists between earthquake wave velocity and shear modulus, which is shown in Equation (1). The shear modulus also determines the propagation velocity of seismic waves in sandy soil. Therefore, when conducting shaking table tests on aeolian sand high embankments, to make the test results more reliable, dynamic shear modulus is a significant parameter for the model design.

$$V_s = \sqrt{\frac{G}{\rho}} \tag{1}$$

where $G$ represents the shear modulus and $\rho$ represents the initial density of the sand.

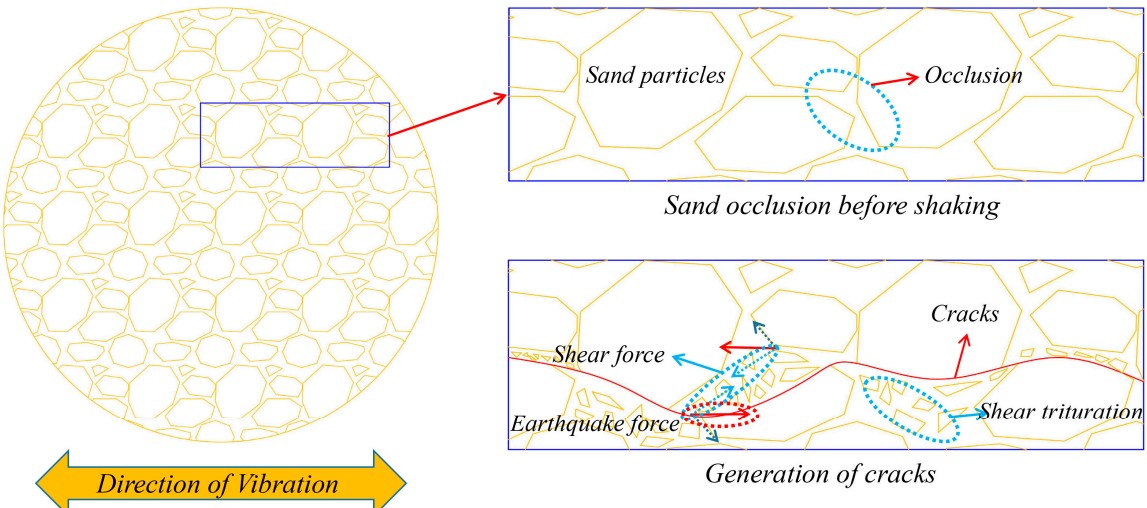

**Figure 2.** Failure mechanism of prototype sand under horizontal accelerations.

## 2.2. Similarity Laws and Materials

The dimensions of the shaking table are 2.2 × 2 m (length × width). Its bearing capacity is 40 kN with a loading frequency range of 0.01–500 Hz. The waveforms of the shaking table can be designed as sinusoidal waves, random waves, and earthquake excitations. The maximum acceleration and amplitude is 27.7 m/s$^2$ and 100 mm, respectively. Shaking table test models were designed according to a prototype of a aeolian sand high embankment—with a compaction degree of 96%—which are located at the northern part of the Mu Us Desert. Based on the dimension of prototype and the carrying capacity of the shaking table, a model size similarity ratio of 1:30 was chosen. The similarity law for the aeolian sand high embankment can be deduced based on the Buckingham π theory [39–41]. Considering the importance of the dynamic shear modulus to the horizontal dynamic response of Aeolian sand embankment as mentioned above, it is necessary to determine the precise dynamic shear modulus constant before making the model. Hardin [42] proposed an empirical formula as shown in Equation (2) to calculate the maximum dynamic shear modulus. According to this empirical formula, the similarity constant of dynamic shear modulus can be calculated as shown in Equation (3) for this research. To select a suitable model sand, a series of dynamic triaxial tests were conducted. The test samples were prepared by adding different contents of quartz sand (30% and 50%) into prototype sand under different degrees of compaction (90% and 86%) using vacuum saturation and isobaric consolidation. The design of dynamic triaxial tests and test results are shown in Figure 3.

$$G_{max} = AP_a(\sigma'/P_a)^{1/2}, \ A = 625\frac{OCR^k}{0.3 + 0.7e^2} \tag{2}$$

where $P_a$ represents the atmospheric pressure, $\sigma'$ represents the initial effective consolidation stress, OCR represents the overconsolidation ratio, and $e$ represents the void ratio.

$$C_G = C_\rho^{1/2}C_l^{1/2} \ C_l = 30 \tag{3}$$

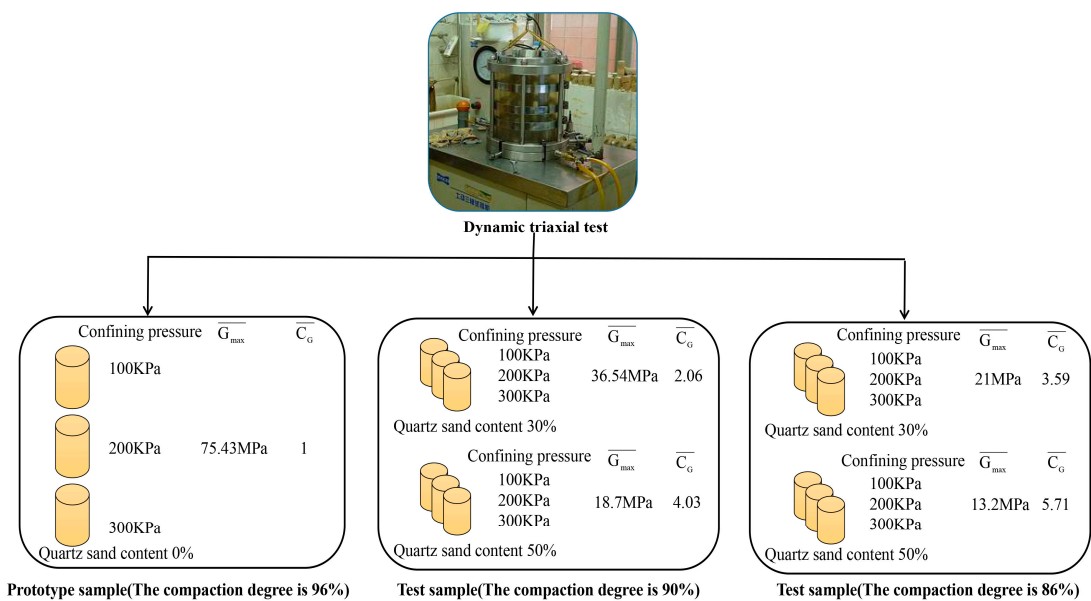

**Figure 3.** Design of dynamic triaxial tests and test results.

The results show that the average dynamic shear modulus similarity constant $\overline{C_G}$ of the specimens (after adding 50% quartz sand with a compaction of 86%) is 5.71. The density similarity constant $C_\rho$ of them is 1.02 according to laboratory tests. As above mentioned, $C_l = 30$. Then, according to Equation (3), $C_G$ is calculated as 5.48. It can be seen that $C_G$ is close to $\overline{C_G}$(5.71). Therefore, sand with a compaction of 86% and a quartz sand content of 50% was determined to be the model material. Using the geometry size, sand density, and gravity acceleration as control variables, the similarity laws of aeolian sand high embankment slope are shown in Table 1.

**Table 1.** Similar laws of aeolian sand high embankment slope.

| Physical Quantity | Similarity Law | Similarity Constants | Remark |
|---|---|---|---|
| Geometry (l) | $C_l$ | 30 | Control variable |
| Mass density (ρ) | $C_\rho$ | 1.02 | Control variable |
| Gravity acceleration (g) | $C_g$ | 1.0 | Control variable |
| Stress (σ) | $C_\sigma = C_\rho C_l$ | 30.6 | |
| Strain (ε) | $C_\varepsilon = C_\gamma = C_\rho^{1/2} C_l^{1/2}$ | 5.53 | |
| Shear modulus (G) | $C_G = C_\rho^{1/2} C_l^{1/2}$ | 5.53 | |
| Displacement (u) | $C_u = C_l C_\gamma = C_\rho^{1/2} C_l^{3/2}$ | 165.95 | |
| Velocity (v) | $C_v = C_\rho^{1/2} C_l^{3/2} C_t^{-1} = C_\rho^{1/4} C_l^{3/4}$ | 12.88 | |
| Input acceleration (a) | $C_a$ | 1.0 | Input control |
| Time (t) | $C_t = C_\rho^{1/4} C_l^{3/4}$ | 12.88 | Input control |
| Frequency (f) | $C_f = C_\rho^{-1/4} C_l^{-3/4}$ | 1/12.88 | |
| Damping ratio (ξ) | $C_\xi$ | 1.0 | |
| Internal friction angle (φ) | $C_\varphi$ | 1.0 | |

## 2.3. Test Cases

Shaking table tests were conducted on three embankment slope models (i.e., embankments with slopes of 1/1.5, 1/1.2, and 1/0.8). Three models were named model 1, model 2, and model 3, respectively. Model 1 was placed in a model box separately, while model 2 and model 3 shared a model box. The dimension of the outside edge of the box is 1960 × 1700 × 800 mm (length × width × height).

The internal size of the box is 1820 × 1500 × 800 mm (length × width × height) without box thickness and boundary treatment materials. The surfaces of the model box parallel to the direction of vibration are transparent. A polystyrene foam board was set behind the model to reduce the box-effect during the test. The experimental setup design is shown in Figure 4.

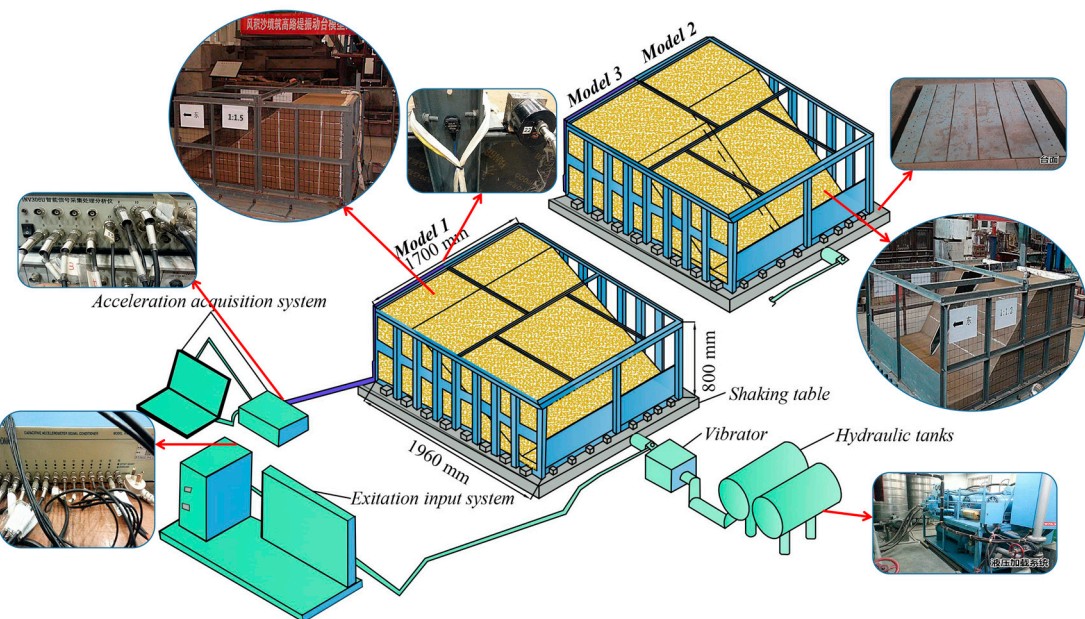

**Figure 4.** Experimental setup design.

The thickness of the sand foundation was 50 mm, and the height of the model was 700 mm according to the similarity theory. The sand was compacted in layers (100 mm/layer) by controlling the compaction degree based on the similarity laws. The construction process of the models is shown in Figure 5.

To observe the shift of the model, 50 × 50 mm squares were drawn at the transparent side of the model box and white trace particles were set at the intersections of the squares. The original positions of white trace particles were marked with a red point. Horizontal accelerometers were used in the shaking table test. The accelerometers set on the vibration table were numbered AC1 and AC3. The accelerometers at the top of the model box were numbered as AC2 and AC4. The accelerometers in slope surfaces were upwardly numbered as A1–A4, A8–A11, and A15–A18. Inner horizontal accelerometers were downwardly numbered as A5–A7, A12–A14, and A19–A21. The accelerometers at the boundary were downwardly numbered as AB1–AB3, AB4–AB6, and AB7–AB9 to test boundary effect, as shown in Figure 6.

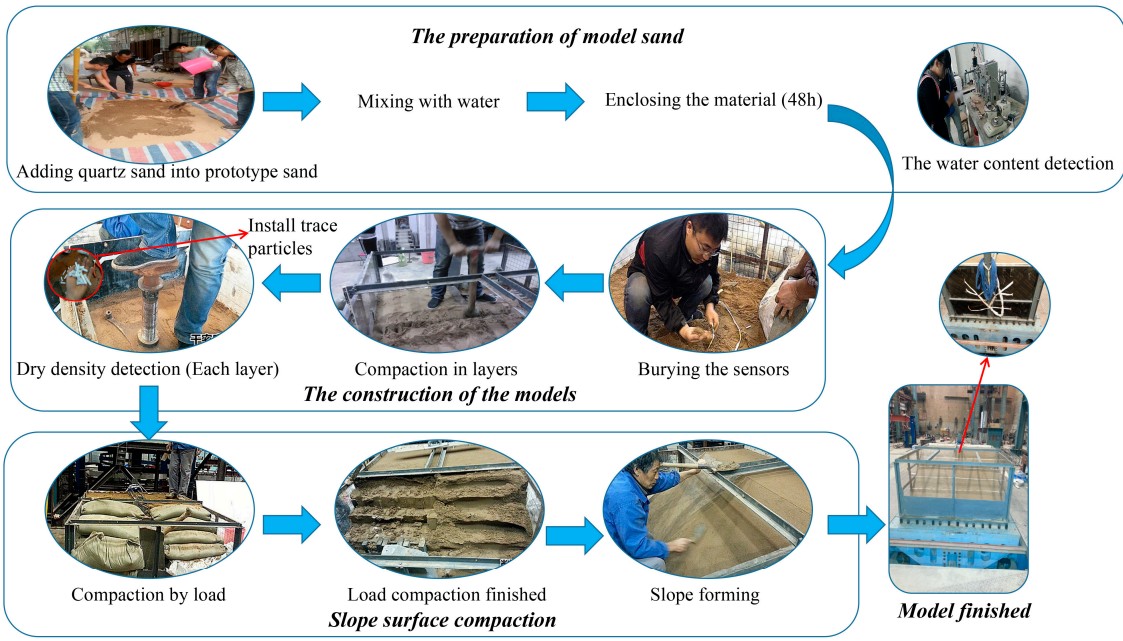

**Figure 5.** Construction process of the models.

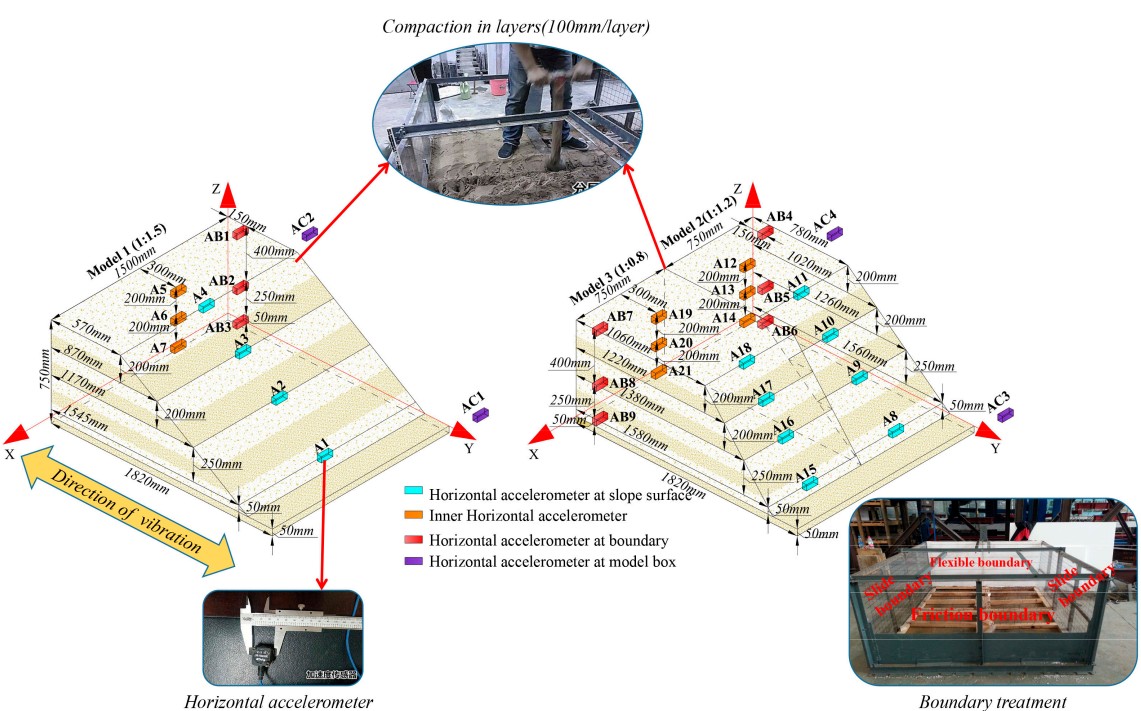

**Figure 6.** Embankment slope models and instruments layout.

## 2.4. Boundary Effect and Loading Method

There are three boundary treatment measures in shaking table tests, which are flexible boundary, sliding boundary, and friction boundary as shown in Figure 6 [43]. The friction boundary was applied to the bottom of the box, which can prevent relative movement.

A sliding boundary was applied to the transparent side of the box. A flexible boundary was applied to the back of the box to reduce the box-effect. In this work, the effect of the frictional boundary was analyzed by comparing the acceleration time–history curves of AC1 and AC3 with those of AB3, AB6, and AB9, the effect of the flexible boundary was analyzed by comparing the

acceleration time-history curves of AB1–AB2, AB4–AB5, and AB7–AB8 with those of A5–A7, A12–A14, and A19–A21. The natural frequency of the box differs from that of the sand to reduce the 'box-effect'; therefore, the natural frequency of the model box was analyzed via hammering test, and results showed that the natural frequencies of the boxes were 43.97 Hz and 43.66 Hz, respectively. The natural frequency of the model was obtained via spectrum analysis of the white noise scanning, and were 17.5 Hz, 16.6 Hz, and 16.6 Hz, respectively; therefore, the 'box-effect' was negligible and the test results were accurate and reliable. Then, Lan Zhou motions and El-Centro motions were selected for the test. The timelines of these motions were compressed according to the time similarity constant ($C_t = 12.88$). The time history of compressed Lan Zhou motions and El-Centro motions are shown in Figure 7. During the test, El- Centro motions and Lan Zhou motions were input alternately. Finally, sinusoidal waves (4 Hz) were input to disrupt the model. White noise was scanned for a frequency during the test. The loading method is shown in Table 2 in which, El-Centro motion was code-named E, Lan Zhou motion was code-named L, white noise was code-named WN, and sinusoidal wave was code-named S.

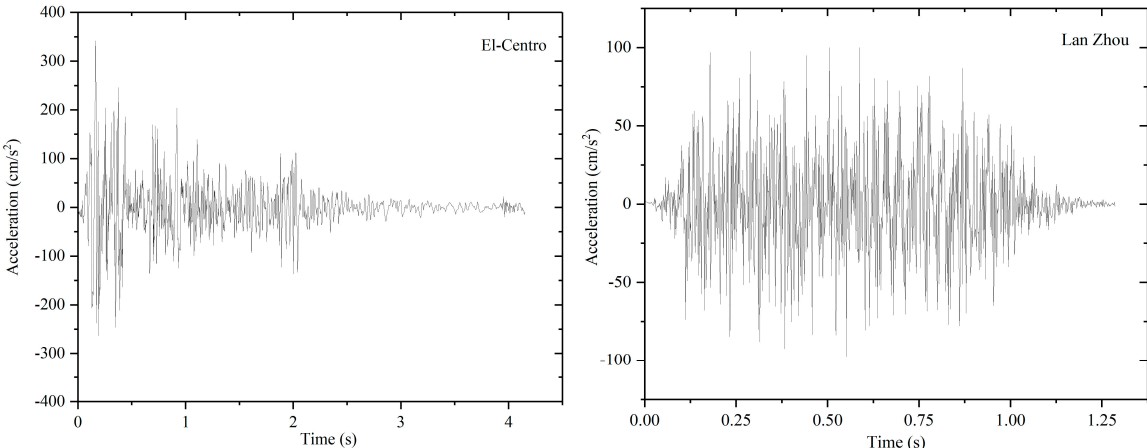

**Figure 7.** Time history of excitations. (El-Centro motion recorded at El Centro, USA, May 18, 1940. M7.1; Lanzhou motion: artificial earthquake wave.).

**Table 2.** Loading scheme of the shaking table tests.

| Serial Number | Seismic Wave | Peak Acceleration/gal(cm/s$^2$) |
|---|---|---|
| 1 | WN1 | 30 |
| 2 | E0.5 | 50 |
| 3 | L0.5 | 50 |
| 4 | WN2 | 30 |
| 5 | E1 | 100 |
| 6 | L1 | 100 |
| 7 | WN3 | 30 |
| 8 | E2 | 200 |
| 9 | L2 | 200 |
| 10 | WN4 | 30 |
| 11 | E3 | 300 |
| 12 | L3 | 300 |
| 13 | WN5 | 30 |
| 14 | E4 | 400 |
| 15 | L4 | 400 |
| 16 | WN6 | 30 |
| 17 | E5 | 500 |
| 18 | L5 | 500 |
| 19 | WN7 | 30 |
| 20 | E6 | 600 |
| 21 | L6 | 600 |
| 22 | WN8 | 30 |
| 23 | E7 | 700 |
| 24 | L7 | 700 |
| 25 | WN9 | 30 |
| 26 | E8 | 800 |
| 27 | L8 | 800 |
| 28 | WN10 | 30 |
| 29 | E9 | 900 |
| 30 | L9 | 900 |
| 31 | WN11 | 30 |
| 32 | E10 | 1000 |
| 33 | L10 | 1000 |
| 34 | WN12 | 30 |
| 35 | S1 (4 Hz) | 1200 |
| 36 | WN13 | 30 |
| 37 | S2 (4 Hz) | 1500 |

*2.5. Rationality of the Earthquake Excitations*

Seismic response laws can be influenced by the value change between the predominant frequency of earthquake excitations and natural frequency of embankment models. More concretely, when conducting shaking table tests, whether or not the predominant frequencies of earthquake excitations are larger than natural frequencies of embankment models may result in a difference of acceleration magnification of test points (becoming greater or smaller). In addition, when the predominant frequency of earthquake excitations and natural frequency of embankment models are close to each other, a resonance phenomenon will happen, which may cause the exaggeration of the acceleration response of models. Therefore, to obtain reliable seismic response results, a constant size relation between the predominant frequency of earthquake excitations and natural frequency of embankment models should be ensured, and the resonance phenomenon should be avoided. Therefore, it is necessary to test the rationality of the chosen earthquake excitations. So in this study, under the loading condition of WN8, the white noise spectral analyses of three models were conducted, and the analysis results of test points (A6–A7; A13–14; A20–A21) are shown in Figure 8, which shows that the natural frequency of three models is 17.5 Hz (1/1.5), 16.6 Hz (1/1.2), and 16.6 Hz (1/0.8),

respectively. Then, according to Fourier spectrum of El-Centro motions and Lan Zhou motions, which is shown in Figure 9, it can be seen that the predominant frequency of these two motions (over 50 Hz) always greater than the natural frequency of three models. Accordingly, the influence of predominant frequency of these two motions was removed, and the material of models and the microcosmic shear failure process during tests were the only factors that influenced seismic response laws. In addition, the maximum input peak acceleration of sinusoidal waves is greater than that of other excitations in this shaking table system, which makes the disruption of models easier, and the resonance phenomenon will not happen under sinusoidal waves due to their low frequency. Therefore, sinusoidal waves (4 Hz) were adopted at the stage of the disruption of models. Generally, the rationality of earthquake excitations and the reliability of seismic response laws were proven according to the above discussion.

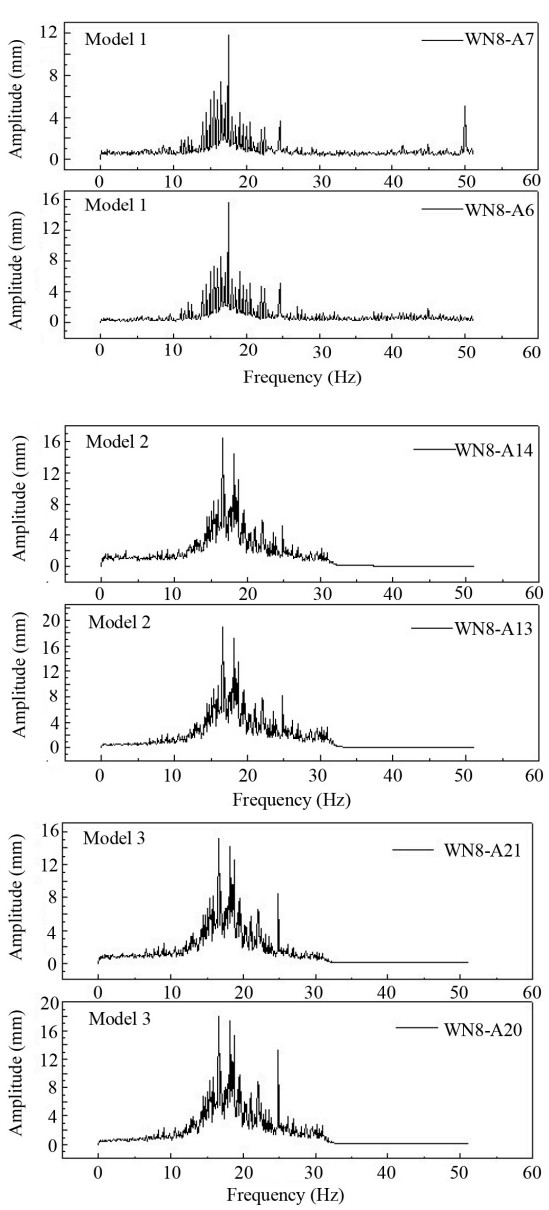

**Figure 8.** White noise spectral analysis of three models.

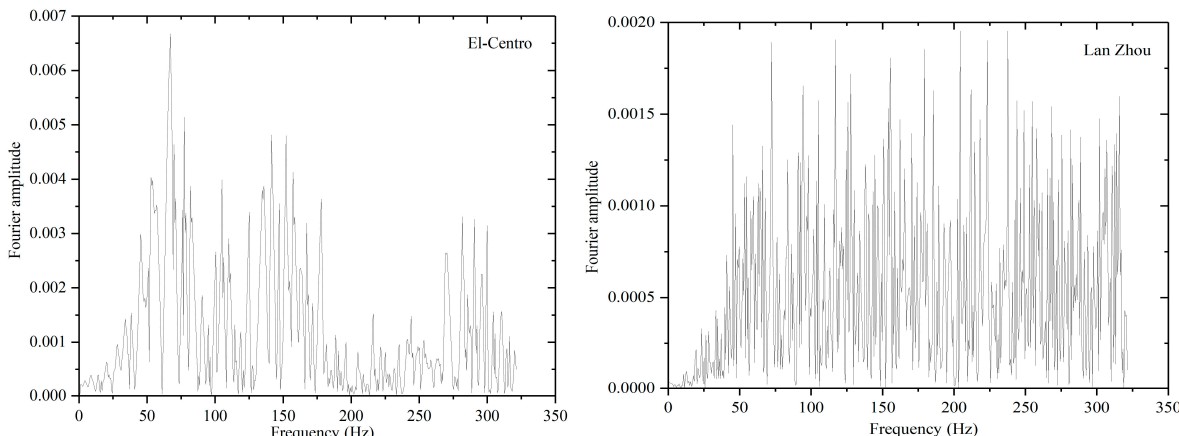

**Figure 9.** Fourier spectrum of excitations.

## 3. Horizontal Acceleration Response Results and Discussion

Horizontal acceleration magnification (HAM) is widely used to study the seismic response of embankment slopes [44]. In this research, acceleration magnification is defined as the ratio of peak acceleration of each test point in slopes to the peak acceleration of the test points on the shaking table (AC1 and AC3)

The distribution of the HAM of four accelerators of the slope surface and that of three accelerators inside the slope were summarized and shown in Figure 10. The results show that the HAM of three models always exceeds 1.0, showing an increasing trend with height. The amplification effects due to the dynamic response of the embankment triggered by both incident and reflected waves are the main reasons for these phenomena. When considering the actual tests, these phenomena can also be explained by the reflection of earthquake waves between sand particles, and the superposition of incident waves and reflected waves along the slope surface. In addition, The HAM of model 1 increases with increasing input peak acceleration. The HAM of model 2 and model 3 shows attenuation when the input peak acceleration reaches a specific value (300 gal in this test). These phenomena can be explained by the failure mechanism of sand under horizontal accelerations. As shown in Figure 11, the destruction of aeolian sand high embankments under earthquakes follows three stages: the reflected wave emergence (RWE) stage, the reflected wave strengthening (RWS) stage, and the acceleration magnification attenuation (AMA) stage. At the RWE stage, the shear trituration of sand particles is not present because the seismic load is sufficiently small; however, the reflection of earthquake waves among sand particles results in the amplification of the input peak acceleration. At the RWS stage, with increasing seismic load, shear trituration of sand particles appeared. These micro-particles, generated by shear trituration, can strengthen the reflection of seismic waves, and this area is named the RWS zone in this paper. At the same time, microcracks appeared as well. Although these microcracks cause slight movement of sand particles, which result in the release of earthquake force and attenuate the HAM to a certain extent, the RWS zone expands rapidly and predominates the change of HAM. Therefore, the HAM increases during this stage. At the AWA stage, with the expansion of the RWS zone, the microcracks connect, and cracks that extend to the slope surface form. Then, the sand near the cracks will shift and permanent deformation will appear at the slope surface, which will lead to a large degree of seismic force release. Although the RWS zone still exists, the cracks predominate the change of HAM. Therefore, the HAM shows attenuation at this stage. According to the distribution of HAM, it can be seen that 300 gal(cm/s$^2$) is the critical acceleration of model 2 and model 3 for the transformation from the RWS stage to the AMA stage. Model 1 only experienced the RWE and RWS stages during the test. These results are highly consistent regardless of whether tests are conducted under El-Centro motions or under Lanzhou motions. Generally, the critical acceleration of embankments during their transformation from the RWS stage to the AMA stage is a reference for the

quantitative estimation of the anti-seismic ability of aeolian sand high embankments. With the increase of this critical acceleration, the anti-seismic ability of aeolian sand high embankments will improve.

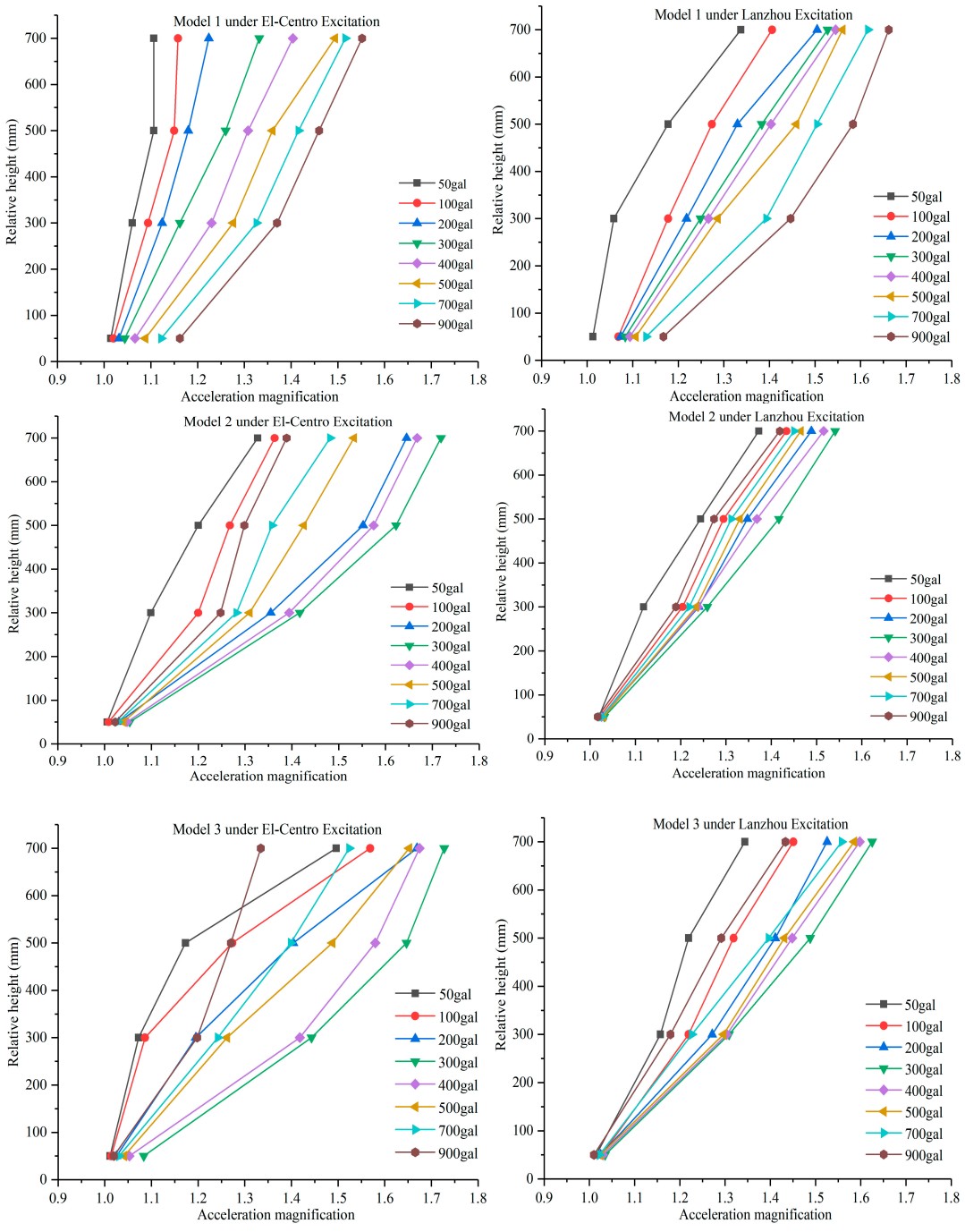

(**a**) Distribution of HAM of the slope surface

**Figure 10.** *Cont.*

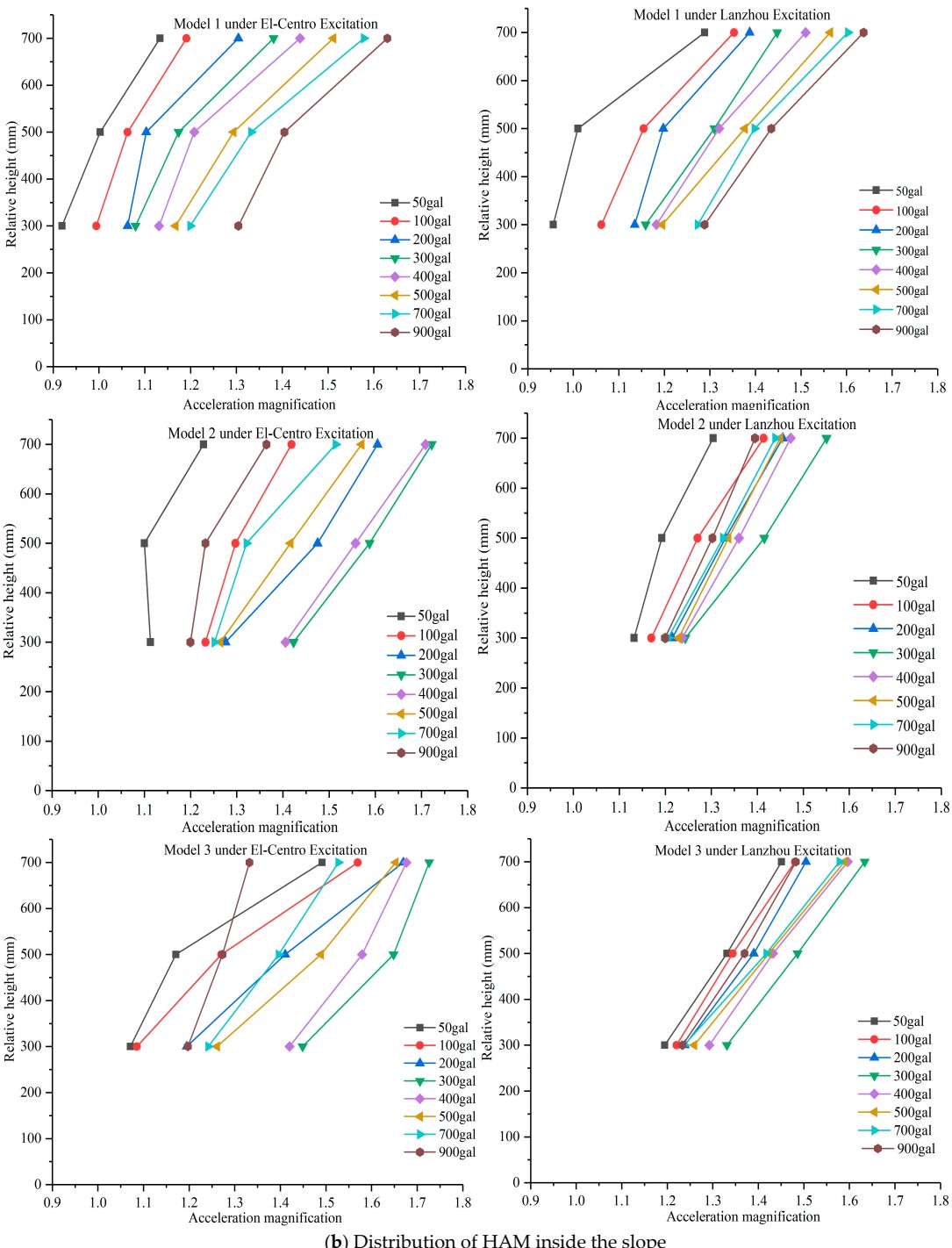

(**b**) Distribution of HAM inside the slope

**Figure 10.** Distribution of HAM (1 gal = 1 cm/s$^2$): (**a**) slope surface (**b**) inside the slope.

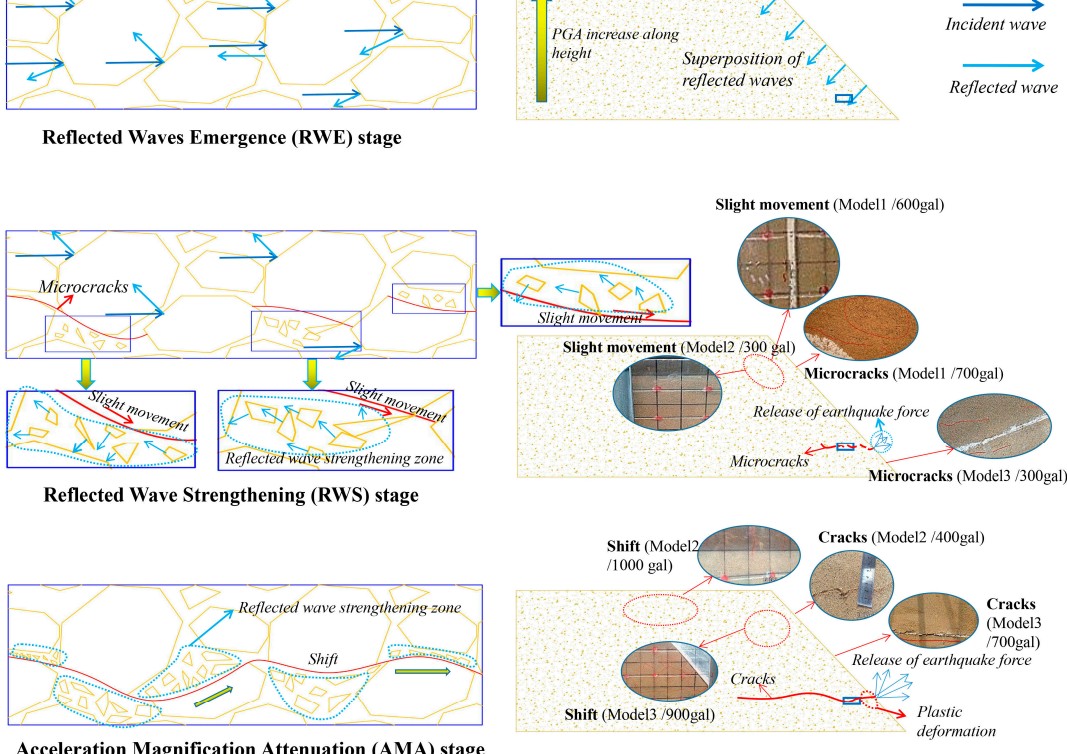

**Figure 11.** Three stages of destruction in shaking table tests.

Taking model 2 under El-Centro motions and model 3 under Lanzhou motions as examples, the relationship between HAM and the input peak acceleration is shown in Figure 12. It is apparent that three stages happened on these two models with increasing input peak acceleration

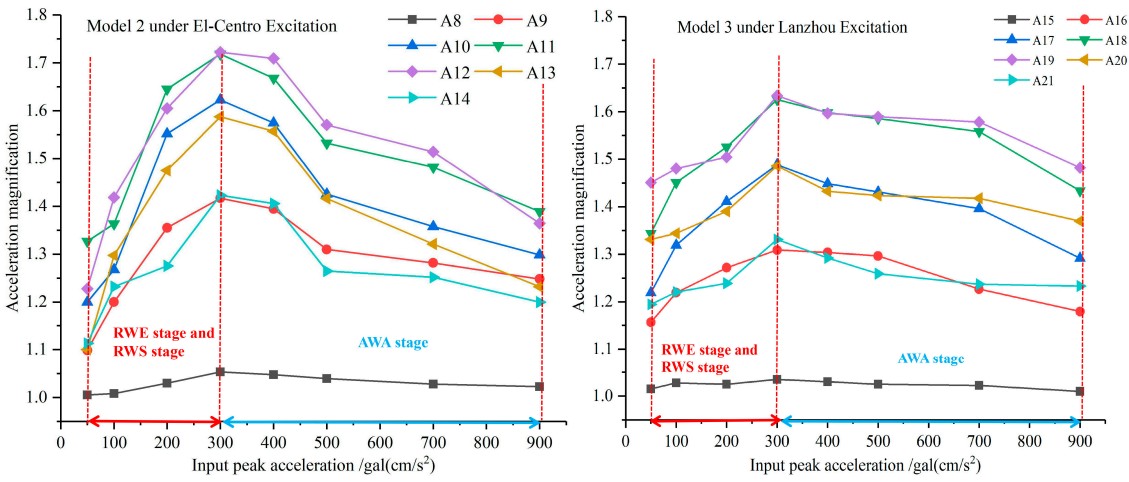

**Figure 12.** Relationship between HAM and input peak acceleration.

Then, taking the critical value (300 gal) of model 2 and model 3 during the transformation from the RWS stage to the AWA stage, the HAM in the slope surface and inside the slope of three models were compared, as shown in Table 3. The average HAM differences of model 1 under El-Centro motions and Lanzhou motions are 0.0723 and 0.081, respectively, which exceed those of model 2 and model 3. The reason for this phenomenon is that when the input peak acceleration reached 300 gal, cracks that extended to the slope surface were formed in model 2 and model 3 (AMA stage start), and permanent deformation appeared at the slope surface, which results in the release of earthquake

force. Then, the reflection wave of the slope surface wakened. Therefore, the HAM between the slope surface and inside the slope at the same height were similar. In contrast, only micro cracks appeared in model 1 and the reflection waves of the slope surface were still strong, which results in a HAM of the slope surface larger than that inside the slope. This phenomenon also indicates the rationality of the definition of destruction stages. Moreover, in terms of micro-damage, this phenomenon proved that the shear-trituration speeds of model 2 and model 3 are greater than that of model 1, which indicates that the slope ratio has a significant influence on this speed. Furthermore, it also indicates that the shear-trituration speeds of three models during earthquakes are the main factor that leads to the failure of models.

**Table 3.** Comparison of HAM between slope surface and interior of slope.

|  | Relative Height/mm | Magnifications under E3 Condition | | | | Magnifications under L3 Condition | | | |
|---|---|---|---|---|---|---|---|---|---|
|  |  | Inside | Surface | Difference | Average | Inside | Surface | Difference | Average |
| **Model 1** | 300 | 1.080 | 1.162 | 0.082 | | 1.159 | 1.249 | 0.09 | |
| | 500 | 1.174 | 1.260 | 0.086 | 0.0723 | 1.310 | 1.383 | 0.073 | 0.0810 |
| | 700 | 1.332 | 1.381 | 0.049 | | 1.448 | 1.528 | 0.08 | |
| **Model 2** | 300 | 1.423 | 1.417 | 0.006 | | 1.243 | 1.259 | 0.016 | |
| | 500 | 1.588 | 1.622 | 0.034 | 0.0147 | 1.415 | 1.417 | 0.002 | 0.0090 |
| | 700 | 1.722 | 1.718 | 0.004 | | 1.550 | 1.541 | 0.009 | |
| **Model 3** | 300 | 1.449 | 1.443 | 0.006 | | 1.331 | 1.309 | 0.022 | |
| | 500 | 1.648 | 1.647 | 0.001 | 0.0030 | 1.486 | 1.488 | 0.002 | 0.0110 |
| | 700 | 1.726 | 1.728 | 0.002 | | 1.634 | 1.625 | 0.009 | |

## 4. Failure Mode Analysis of Models

Huang [45] classified the instability mechanism of natural slopes under strong earthquakes into several types, in which tension-shattering sliding mainly occurs in the toppling slopes or in transverse structural slopes. The failure process is that under strong earthquakes, vertical tensile cracks form in the slope. Meanwhile, horizontal cracks, which are generated by tensile shear destruction, also form inside the slope due to horizontal acceleration. Then, the collapse area was formed by the connection of vertical cracks and horizontal cracks, which resulted in tension-shattering sliding. Such landslides are usually characterized by a rough steep sliding surface [46–49].

According to the failure mode of models under sinusoidal waves, it was found that the instability mechanism of model 2 and model 3 belong to the tension-shattering sliding type (Figure 13). Moreover, the instability mechanism of these two models is in accordance with the defined destruction stages. At the AMA stage, horizontal cracks that extend to the slope surface were formed and these cracks connected with the vertical tensile cracks, resulting in tension-shattering sliding. The tension-shattering sliding is dominated by the properties of sand, especially the internal friction angle. The collapse accumulation includes accumulation under strong earthquakes and accumulation under gravity.

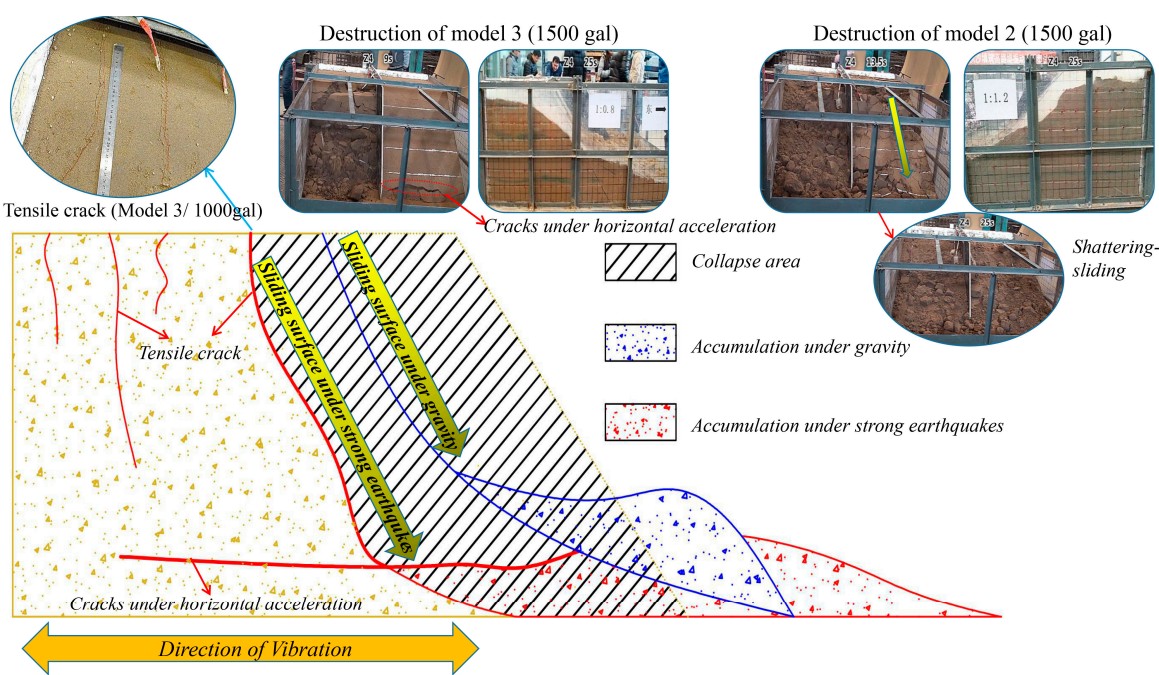

**Figure 13.** Tension-shattering sliding of model 2 and model 3.

Model 1 only shows partial collapse during the test as shown in Figure 14. According to the above-mentioned context, model 1 only experienced the RWE stage and the RWS stage. The micro cracks that were generated at the RWS stage cannot lead to the collapse of model 1, but these micro cracks connect with small tensile cracks at the slope surface; then, the instability of this part happened under gravity, and a partial collapse occurred.

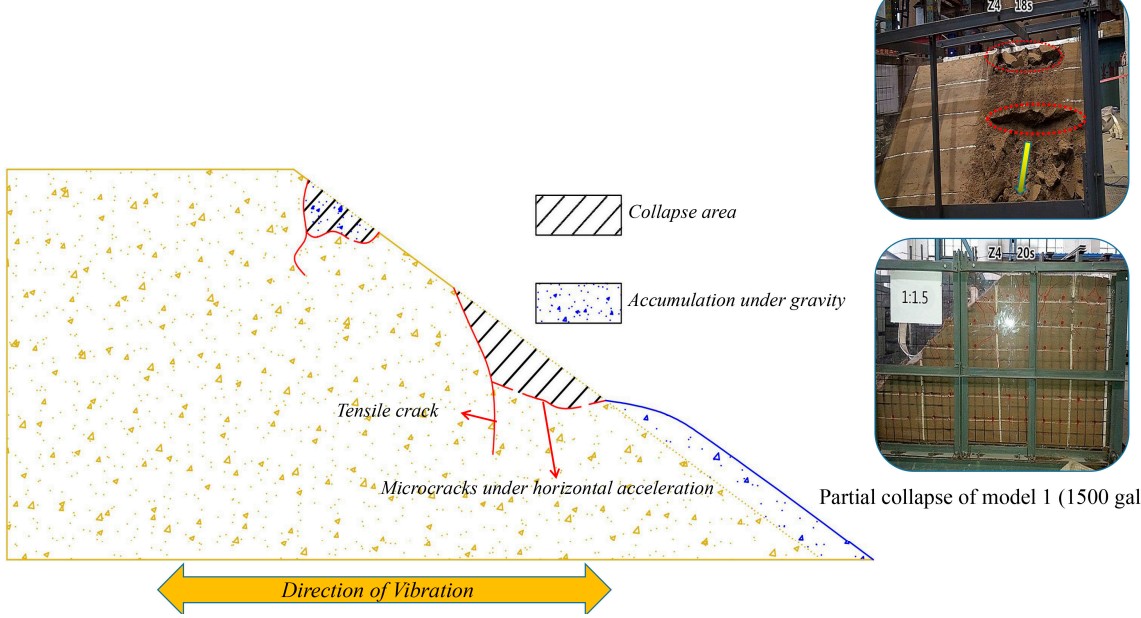

**Figure 14.** Partial collapse of model 1.

On the other hand, it also shows that the slope ratio has a significant influence on the seismic capacity of aeolian sand high embankments.

## 5. Conclusions

In this paper, specimens of aeolian sand high embankments were designed and constructed, complying with similarity laws, and shaking table tests were performed to study the seismic response. The following primary conclusions can be drawn:

(1) The horizontal acceleration magnification (HAM) of the three model embankments always exceeds 1.0, and shows an increasing trend with height. The amplification effects due to the dynamic response of the embankment triggered by both incident and reflected waves are the main reasons for these phenomena.

(2) The HAM of model 1 (1:1.5) increases with increasing input peak acceleration. The HAM of model 2 (1:1.2) and model 3 (1:0.8) shows attenuation when the input peak acceleration reaches a certain threshold value (300 gal in this test)

(3) The microcosmic failure mode of the Aeolian sand embankments can be defined in three stages: the reflected wave emergence (RWE) stage, the reflected wave strengthening (RWS) stage, and the acceleration magnification attenuation (AMA) stage. 300 gal (cm/s$^2$) is the critical acceleration of model 2 and model 3 transform from the RWS stage to the AMA stage. Furthermore, model 1 only experienced the RWE stage and the RWS stage during the test. Therefore, this critical acceleration can provide a significant guide on quantitative estimation of the anti-seismic ability of aeolian sand high embankments.

(4) The embankment instability mechanisms of model 2 and model 3 belong to the type of tension-shattering sliding, and a collapse area formed by the horizontal continuous cracks and vertical tensile cracks. Only partial collapse occurred in model 1. The slope ratio has a significant influence on the seismic capacity of aeolian sand high embankments.

In general, this study shows that the slope ratio exerts a significant influence on the seismic response of aeolian sand high embankments. The critical acceleration of embankments during their transformation from the RWS stage to the AMA stage is a reference for the estimation of the anti-seismic ability of aeolian sand high embankments. The instability mechanism of the models under horizontal acceleration also informed the highly efficient seismic design of aeolian sand high embankments.

**Author Contributions:** Conceptualization, Z.Z., J.L., S.S., and T.L.; Data curation, Z.Z., J.L., S.S., and T.L.; Funding acquisition, Z.Z.; Investigation, J.L.; Methodology, J.L., S.S.; Writing—original draft, Z.Z., J.L., and T.L.

**Funding:** This work is financially supported by the National Key R&D Program of China (no. 2018YFC0808706).

**Conflicts of Interest:** The authors declare there is no conflicts of interest of this regarding the publication of this paper.

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
