# Peer review of "Seismic Response of Aeolian Sand High Embankment Slopes in Shaking Table Tests"

_applsci, doi:10.3390/app9081677_

Round 1
Reviewer 1 Report
The paper is generally well written and organized.
Few minor recommendations:
- in figure 8, use identical axis range values for all the plots
- same comment for figure 10. For comparative purposes, this will enhance the readability of charts
- consider to extend the list of references and include also European and US contributions
Author Response
Point 1: In figure 8, use identical axis range values for all the plots.
Response 1: Thank you very much for the comments of the reviewer. We have already adopted identical axis range values for all the plots in figure 8 in the revised paper.
Point 2: Same comment for figure 10. For comparative purposes, this will enhance the readability of charts.
Response 2: We are so grateful for reviewer’s technical and helpful comments. We have already adopted identical axis range values for all the plots in figure 10 in the revised paper.
Point 3: Consider to extend the list of references and include also European and US contributions.
Response 3: Thanks a million for the reviewer’s comments. We have already realized that it is important to present the contributions of European and US. There is no doubt that many excellent scholars come from these countries, and their studies provide a significant guide on our research. Therefore, after careful reading and selecting, we decided to add some new references from these countries.
ALL DETAILS are shown in PDF.

Reviewer 2 Report
The paper is well written and the findings are justified. The results show that aeolian sand high embankment slopes, depending on the slope ratio, may have significant performance and capacities. Also, critical accelerations can provide a significant guide on quantitative estimation of the anti-seismic ability of aeolian sand high embankments.
Author Response
Point 1: The paper is well written and the findings are justified. The results show that aeolian sand high embankment slopes, depending on the slope ratio, may have significant performance and capacities. Also, critical accelerations can provide a significant guide on quantitative estimation of the anti-seismic ability of aeolian sand high embankments.
Response 1: Thank you very much for your appreciation of our work.

Reviewer 3 Report
This paper reported the seismic response and failure mechanism for model embankments using shaking table tests. The reviewer can praise the authors’ efforts for organizing intensive model tests and describing the results of them. After an additional integration of the incomplete parts like below, the paper can be fully accepted for publication.
- Regarding an issue for scaling factors (similarity laws in the manuscript), the reference [56] in the manuscript cannot be explained the general Buckingham π theorem. The reviewer would suggest reviewing two references below and scrutinise the scaling factors of Table 1 for 1-g shaking table tests in addition to Iai (1989). The two references discussed the scaling factors not only for the N-g dynamic centrifuge tests but also for the comparison to 1-g shaking table test.
(1) S. Iai, T. Tobita, T. Nakahara. Generalized scaling relations for dynamic centrifuge tests. Geotechnique, 2005, 55(5), 355-362, doi:10.1680/geot.2005.55.5.355
(2) Park, H.J.; Kim, D.S. Centrifuge modelling for evaluation of seismic behaviour of stone masonry structure. Soil Dyn. Earthq. Eng. 2013, 53, 187–195, doi:10.1016/j.soildyn.2013.06.010
- The acceleration values were used in gal and m/s2 units. The units could be unified. A few acceleration units would be changed to superscript (m/s2, cm/s2 -> m/s2, cm/s2).
- The frequency of sinusoidal waves could be expressed in Table 2. The relationship between the natural frequency of the embankment slope model and input frequency, or the reason why the excitation of sinusoidal waves could be discussed. Additionally, the frequency contents of two time- domain signals of Figure 7 would be explained.
- The authors concluded the slope ratios (1/1.5, 1/1.2 and 1/0.8) exerted a significant influence. The reviewer wonders if the failure/instability mechanism could be related to material properties of soil, especially, internal friction angle 37.5 degrees.
Author Response
Point 1: Regarding an issue for scaling factors (similarity laws in the manuscript), the reference [56] in the manuscript cannot be explained the general Buckingham π theorem. The reviewer would suggest reviewing two references below and scrutinise the scaling factors of Table 1 for 1-g shaking table tests in addition to Iai (1989). The two references discussed the scaling factors not only for the N-g dynamic centrifuge tests but also for the comparison to 1-g shaking table test.
(1) S. Iai, T. Tobita, T. Nakahara. Generalized scaling relations for dynamic centrifuge tests. Geotechnique, 2005, 55(5), 355-362, doi:10.1680/geot.2005.55.5.355
(2) Park, H.J.; Kim, D.S. Centrifuge modelling for evaluation of seismic behaviour of stone masonry structure. Soil Dyn. Earthq. Eng. 2013, 53, 187–195, doi:10.1016/j.soildyn.2013.06.010
Response 1: Thank you very much for your helpful suggestions. After reading the papers recommended by the reviewer, we scrutinised the scaling factors of Table 1 again, and we found the derivation method of scaling factors in our paper in accordance with the method that introduced by the recommended papers (Scaling factors for 1g tests), and these papers could explain the general Buckingham π theorem in a more detailed way. Furthermore, as we expressed in the paper, we designed a series of triaxial dynamical tests to obtain similar constants of dynamic shear modulus of different model sand. Then, according to these constants, the most suitable model sand was chosen. This innovative idea also in accordance with the content of the first recommended paper, which expresses that “if the stress-dependent behaviour of soil is evaluated for the entire stress–strain range by preliminary laboratory tests, then the scaling factor for strain is directly obtained from those test results.” For above reasons, it is our pleasure to add these two papers into our references.
Point 2: The acceleration values were used in gal and m/s2 units. The units could be unified. A few acceleration units would be changed to superscript (m/s2, cm/s2 -> m/s2, cm/s2).
Response 2: Many thanks for your careful reading. To solve this problem, we changed the unit of the ordinate of Figure 7 (Time history of excitations). All acceleration values are used in cm/s2 (gal) unit now. Furthermore, we are sorry for missing some superscripts, and we have already changed them in the revised paper.
Point 3: The frequency of sinusoidal waves could be expressed in Table 2.
Response 3: Thanks for the reviewer’s comments. The frequency of sinusoidal waves is 4 Hz. We have already expressed it in Table 2.
Point 4: The relationship between the natural frequency of the embankment slope model and input frequency, or the reason why the excitation of sinusoidal waves could be discussed.
Response 4: To address your helpful comments, we added a new subtitle 2.5 Rationality of the Earthquake excitations. In this section, we discussed that the seismic response laws can be influenced by the size relationship change between the predominant frequency (Input frequency) of earthquake excitations and natural frequency of embankment models. When conducting shaking table tests, whether or not the predominant frequency of earthquake excitations are greater than natural frequency of embankment models may results in the increasing or decline of the acceleration magnification. Therefore, to obtain reliable seismic response results, a constant size relation between the predominant frequency of earthquake excitations and natural frequency of embankment models should be ensured. So, after adding some results of the white noise spectral analysis of three models and the Fourier spectrum of El-Centro motions and Lan Zhou motions, we found that the predominant frequency(Input frequency) of these two motions always greater than the natural frequency of three models. Therefore, at the seismic response stage, the acceleration magnification will not be influenced by the predominant frequency of excitations. In this way, we can say that the material of the models and the microcosmic shear failure process during tests were the only factors that influenced seismic response laws. In addition, the maximum input peak acceleration of sinusoidal waves is greater than that of other excitations in this shaking table system, which makes the disrupt of models easier, and the resonance phenomenon will not happen under sinusoidal waves due to its low frequency. Therefore, sinusoidal wave (4 Hz) was adopted at the stage of the disruption of models. Generally, we proved the reliability of our test results again according to this discussion.
Point 5: Additionally, the frequency contents of two time- domain signals of Figure 7 would be explained.
Response 5: Many thanks for the reviewer’s comments. We used the Fast Fourier Transform (FFT) function of MATLAB to obtain the Fourier spectrum of El-Centro motions and Lan Zhou motions. The frequency contents of two time- domain signals can be explained by the Fourier spectrum of excitations, as shown in Figure 9 in the revised paper. In addition, we can provide the command statement of the Fourier transform used in MATLAB if required.
Point 6: The authors concluded the slope ratios (1/1.5, 1/1.2 and 1/0.8) exerted a significant influence. The reviewer wonders if the failure/instability mechanism could be related to material properties of soil, especially, internal friction angle 37.5 degrees.
Response 6:
Thanks for the reviewer’s comments. We are so grateful that you can express your own opinion on the failure mechanism of the Aeolian sand high embankments under horizontal accelerations. There is no doubt that the properties of soil have a significant influence on the seismic capacity of the embankments, and the internal friction angle of soil plays an important role on the slope stability analysis.
When a soil slope begins to slide, the relative movement of soil particles happen. This relative movement must overcome two kinds of friction, the sliding friction and the occlusion friction. Considering the Aeolian sand high embankment slope, as we expressed in the paper, the seismic capability of it mainly depends on the occlusion force between sand particles. Therefore, when earthquakes happen, the occlusion friction is the main factor to prevent the relative movement of Aeolian sand high embankments. Besides, we also explained that the micro failure of aeolian sand under horizontal accelerations can be described as a shear-trituration between sand particles, which leads to the generation of cracks. Moreover, according to the distribution of horizontal acceleration magnification (HAM) of three models and the destruction stages definition, we found the shear-trituration speeds of model 2 and model 3 are greater than that of model 1. For example, cracks that extent to the slope surface were formed in model 2 and model 3 when input peak acceleration reached 300 gal. In contrast, only micro cracks appeared in model 1. At the final stage of excitation loading, the shear-trituration speed difference leads to the collapse of model 2 and model 3 (cracks formed), but only part collapse of model 1. Therefore, we think that the shear-trituration speeds of three models during earthquakes are the main factor that result in the failure of the models, and the slope ratio has a significant influence on this speed (The different reflection waves’ property along the different ratio of the surface may results in the shear-trituration speeds difference. Actually, this interesting phenomenon encourages us to do more research in depth in the future).
However, we completely agree that the properties of the soil, especially, internal friction angle, dominates the failure of the models when the horizontal cracks connect with the tensile cracks. Nevertheless, considering that the actual research purpose is to find the seismic capacity estimation method of Aeolian sand high embankment slopes, few research values will exists when the the steep sliding surface form and severe deformation of embankments happens.
For the above reasons, we decided to add some contents in the revised paper to explain the influence of slope ratio on the damage of models more intuitive. Also, to enhance the serious of the paper, we added a sentence to emphasize that the properties of the soil, especially, internal friction angle, dominates the failure of the models when the horizontal cracks connect with the tensile cracks.
All the details are in PDF
